# Effects of Germination on the Structure, Functional Properties, and In Vitro Digestibility of a Black Bean (*Glycine max* (L.) Merr.) Protein Isolate

**DOI:** 10.3390/foods13030488

**Published:** 2024-02-03

**Authors:** Xin-Hui Wang, Zhen-Jia Tai, Xue-Jian Song, Zhi-Jiang Li, Dong-Jie Zhang

**Affiliations:** 1College of Food, Heilongjiang Bayi Agricultural University, Xinfeng Road 5, Daqing 163319, China; w604466213@163.com (X.-H.W.); tzj970802@163.com (Z.-J.T.); byndsxj@126.com (X.-J.S.); lizhijiang@126.com (Z.-J.L.); 2National Coarse Cereals Engineering Research Center, Daqing 163319, China; 3Key Laboratory of Agro-Products Processing and Quality Safety of Heilongjiang Province, Daqing 163319, China

**Keywords:** black bean protein isolate, germination, structural characteristics, functional properties, in vitro digestibility

## Abstract

The utilization of black beans as a protein-rich ingredient presents remarkable prospects in the protein food industry. The objective of this study was to assess the impact of germination treatment on the physicochemical, structural, and functional characteristics of a black bean protein isolate. The findings indicate that germination resulted in an increase in both the total and soluble protein contents of black beans, while SDS-PAGE demonstrated an increase in the proportion of 11S and 7S globulin subunits. After germination, the particle size of the black bean protein isolate decreased in the solution, while the absolute value of the zeta potential increased. The above results show that the stability of the solution was improved. The contents of β-sheet and β-turn gradually decreased, while the content of α-helix increased, and the fluorescence spectrum of the black bean protein isolate showed a red shift phenomenon, indicating that the structure of the protein isolate and its polypeptide chain were prolonged, and the foaming property, emulsification property and in vitro digestibility were significantly improved after germination. Therefore, germination not only improves functional properties, but also nutritional content.

## 1. Introduction

With the continuous growth of the global population, in addition to changes in social demographics and other contributing factors, there is an increasingly pervasive demand for resources among individuals. Economic development and the advancement of urbanization are driving a significant transformation in human dietary patterns. There is an increasing recognition among individuals regarding the pivotal role of protein in maintaining a healthy diet, which has led to a growing demand for protein. Traditional sources of animal protein, such as animal husbandry and fisheries, have a significant environmental impact while also facing limitations in terms of nutrition and price [1]. The livestock farming industry has significant environmental implications, exerting adverse effects on soil quality, water resources, and air pollution [2]. For instance, the supply of animal feed and the discharge of animal manure contribute to land degradation and the contamination of groundwater [3].

At the same time, improper waste disposal can result in the contamination of nearby water bodies and an increase in greenhouse gas emissions such as carbon dioxide and methane into the atmosphere, contributing to global climate change and other associated issues [4]. Taking into account nutritional recommendations, the environmental costs of food production and consumption, and adaptations to local social and economic circumstances, some countries have integrated aspects of environmental sustainability into their food-based dietary guidelines [5,6]. Thus, protein consumption patterns will gradually shift toward plant-derived sources that are more environmentally sustainable. Currently, the global protein supply is dominated by plant-based sources (57%), with meat accounting for 18%, dairy products for 10%, fish and shellfish for 6%, and other animal products for 9% [7]. The utilization of plant-derived proteins offers numerous advantages, including their ability to fulfill the protein requirements of individuals without imposing excessive strain on the environment. This contributes to the promotion of ecological conservation and sustainable development [8]. The incorporation of plant-derived proteins also contributes to the amelioration of chronic diseases, such as obesity and cardiovascular disease, thereby promoting overall health. Beans, such as soybeans, black beans, and vegetable beans, are a sustainable source high-quality proteins that contain various amino acids essential for the human body, particularly lysine and isoleucine [9,10]. Replete with protein, indispensable amino acids, anthocyanins, isoflavones, and polyunsaturated fatty acids, black beans are extensively cultivated and consumed worldwide. The bioactive compounds present in BS exhibit antioxidant, anti-cancerous, anti-diabetic, anti-obesity, anti-inflammatory, and cardio- and neuroprotective activities [11,12]. Moreover, black beans exhibit a higher protein content compared to other soybean species and are rich in essential amino acids, making them an exceptional source for protein extraction and modification [13]. Black bean protein isolates exhibit excellent solubility, emulsification, emulsion stability, and antioxidant properties, making them highly valuable for applications in the food industry [14]. However, to date, the majority of studies pertaining to black beans have predominantly focused on analyzing anthocyanins and phenolics present in the seed coat, while insufficient attention has been paid to exploring its protein components [15].

The process of germination has been widely acknowledged as an environmentally sustainable, cost-effective, and efficient approach for the accumulation of bioactive substances, thereby exerting a positive influence on the nutritional profile of cereals [16]. Germination can enhance the metabolism of seeds, leading to the decomposition and breakdown of nutrients and anti-nutrients in seeds, as well as promoting the synthesis of secondary metabolites from nutrients, thereby improving the nutritional value of germinated seeds [17]. Beginning in the 1980s, the consumption of germinated seeds also became popular in various countries due to consumer demand for nutrition and foreign health foods. In fact, the majority of studies that focus on black beans primarily address the augmentation of the content and activity of phenolic compounds, and there is a scarcity of research reporting on the impact of germination on protein isolates derived from black beans [18,19]. Therefore, the objective of this study was to investigate the impact of germination on a black bean protein isolate and explore its electrophoretic characteristics and changes in its amino acid content, protein structure, solubility, and emulsification properties. This study contributes novel research ideas and directions for the processing and application of black bean protein isolates.

## 2. Materials and Methods

### 2.1. Materials and Main Reagents

Black beans were purchased from Beidahuang Agricultural Products Co. An electrophoretic reagent was purchased from Beijing Solarbio Technology Co., Ltd. (Beijing, China). α-Amylase (40,000 U/g), pepsin (250 U/mg), trypsin (250 U/mg), and sodium taurocholate were purchased from Solarbio Technology Co., Ltd. Ultrapure water was used for all experiments.

### 2.2. Instruments

The following instruments were used: a Fluoromax-4 fluorescence spectrometer (HITACHI, Tokyo, Japan); a SevenEasy pH meter and an EL 204 electronic balance (Mettler-Toldo Instruments Co., Ltd., Shanghai, China); a frozen centrifuge (Sigma-Aldrich Company, St. Louis, MO, USA); a UV 5300PC UV-Visible Spectrophotometer (Shanghai Analysis Instrument Co., Ltd., Shanghai, China); and a TENSOR II Fourier transform infrared spectrometer (Bruker (Beijing) Scientific Technology Co., Ltd., Beijing, China).

### 2.3. Germination Treatment and Preparation of Black Bean Protein Isolate

Black beans (*Glycine max* (L.) merr.) without any moth damage or epidermal damage were selected. They were cleaned and soaked in a 10% sodium hypochlorite solution for 5 min, and they were then rinsed with distilled water to remove any residual sodium hypochlorite on the surface. The cleaned black beans were soaked in distilled water at a ratio of 1:5 (*w*/*v*) for 9 h at room temperature (25 °C), with the water changed 1 to 2 times during this period. The germination process was carried out in an automatized germination chamber (CB-A360B, Foshan, China) in the dark at 25 °C with 10 s of irrigation every 1 h. The germinated seeds were harvested after 0, 24, 48, 72, and 96 h, referred to as R, G0, G24 G72, G76. The samples were dried at 45 °C in an air-blast drying oven, finely ground into powder, and sieved through an 80-mesh sieve. The black bean protein isolate was extracted from the germinated black bean powder according to the method described by Zheng et al. [20]. Briefly, germinated black bean powder was mixed with petroleum ether at a ratio of 1:5 and stirred at room temperature for 3 h. The petroleum ether was discarded and the precipitate was dried to obtain germinated black bean defatted powder. The germinated black bean defatted powder was mixed with distilled water at a ratio of 1:10, the pH of the solution was adjusted to 8.0 with 1 M NaOH, and after stirring for l.5 h at 50 °C, it was centrifuged for 20 min at 4 °C and 6000× *g*. The supernatant was adjusted to pH 4.0 with 1 M HCl. The supernatant was centrifuged at 4 °C for 20 min under the condition of 6000× *g*. The precipitate obtained from the centrifugation was washed twice, and then freeze-dried to obtain the germinated black bean isolate protein, which was stored in the refrigerator at −80 °C.

### 2.4. Determination of Total Protein Content, Soluble Protein Content, and Protein Solubility in Germinated Black Bean Powder

The protein content of the germinated black bean powder was determined using the Kjeldahl nitrogen determination method with modifications based on Ma’s protocol [21]. The soluble protein content of the germinated black bean powder and the solubility of the protein isolated from germinated black beans were determined following the method described by Ma et al. [21], with slight modifications. Bovine serum albumin was used as a standard, and a standard curve equation was established as y = 2.604x + 0.0103 (R^2^ = 0.9989).

### 2.5. Amino Acid Composition Analysis of the Germinated Black Soya Bean Isolated Protein

The amino acid composition of the isolated protein was determined according to the method reported by Mokni et al. [22]. Protein isolate samples were subjected to hydrolysis using 6 mol/L hydrochloric acid at a temperature of 110 ± 1 °C for a duration of 22 h. Subsequently, the resulting hydrolysate was evaporated to dryness and reconstituted using 1–2 mL of citrate buffer (pH 2.2). The reconstituted samples were then filtered through a 0.22 μm filter membrane to obtain the test solution. Amino acid fractions were subsequently analyzed using ion exchange chromatography, and the results were expressed with units of g/100 g.

### 2.6. Protein Structure Determination

#### 2.6.1. Polyacrylamide Gel Electrophoresis (SDS-PAGE)

The protein sample was dissolved in a 0.1 mol/L NaOH solution, resulting in a protein concentration of 0.5 mg/mL. Subsequently, an equal volume of loading buffer was added to the solution, followed by the thermal denaturation of the protein in a boiling water bath for 5–8 min. Electrophoresis was conducted using a 5% concentrated gel, a 12% separated gel, and 15 µL of sample loading buffer. Electrophoresis was terminated when the bromophenol blue indicator band reached a distance of 5 mm from the bottom edge of the separation glue. After 2 h of dyeing, the samples were eluted with a decolorizing solution for 2 h. Subsequently, an electrophoresis analysis using Image J (win-64) software enabled the determination of the relative content for each subunit [23].

#### 2.6.2. Fourier Infrared Spectrum Analysis

The infrared spectrum of the germinated black bean protein was recorded at 25 °C using a TENSOR II Fourier transform infrared spectrometer (Bruker, Billerica, MA, USA). The sample was homogenized and pulverized with potassium bromide at a ratio of 1:100; it was then compressed into thin slices using a hydraulic press. Full band scanning was conducted between 4000 and 400 cm^−1^. The average spectral scan was 32, with a resolution of 4 cm^−1^. Peak-Fit v4.12 software was employed to fit the protein amide I region data (1600–1700 cm^−1^), enabling the determination of relative content pertaining to the protein’s secondary structure [24].

#### 2.6.3. Fluorescence Spectrometry

Fluorescence dynamics were recorded using a FluoroMax-4CP fluorescence spectrometer. The protein concentration was set at 0.5 mg/mL. The conditions for fluorescence spectral determination were as follows: quartz cuvette light path = 1 cm; excitation wavelength = 280 nm; emission wavelength = 300–500 nm; excitation and emission slit width = 5 nm; electric tension = 700 mV; and scan speed = 200 nm/min [25].

#### 2.6.4. Ultraviolet Spectrometric Measurements

A 1 mg/mL solution of germinated black bean protein isolate was prepared using a phosphoric acid buffer (0.01 mol/L, pH 7.0). After centrifugation at 4000 r/min for 10 min, the supernatant was collected for UV spectrum scanning within the range of 200–350 nm.

#### 2.6.5. Particle Size Distribution and ζ Potential

The particle size of the black bean was observed using a transmission electron microscope. The germinated black bean protein solution was adsorbed onto a carbon-coated grid, followed by two washes with double-distilled water. Subsequently, the sample was negatively stained using 2% uranyl acetate and imaged using an JEM-2100Plus transmission electron microscope (JEOL (Beijing) Co., Ltd., Beijing, China) [26]. The zeta potential measurement was obtained utilizing a PALS-Zeta potentiometer. A sample solution with a concentration of 0.5 mg/mL was meticulously prepared in a PBS buffer (50 mmol/L, pH 7.0), comprising a total volume of 1 mL, and subjected to analysis at an ambient temperature of 25 °C.

### 2.7. Turbidity

The turbidity of each germinated black bean protein isolate sample was measured using a spectrophotometer (Lambda 1050 UV/VIS/NIR Spectrometer, PerkinElmer, Waltham, MA, USA). The protein concentration was 1 mg/mL, and DI water was used as the blank. The absorbance at 600 nm of each sample represented the turbidity [27].

### 2.8. Foaming Ability Measurements

Foaming capacity (FC) and foam stability (FS) were assessed in accordance with the methodology outlined in [28]. Foam was obtained by homogenizing 15 mL of protein solution (5 mg/mL) for 1 min at 10,000 rpm using a T18BS25 homogenizer (IKA (Guangzhou) Instrument Co., Ltd., Guangzhou, China). The foam volume (V_0_) was immediately measured, and the foam volume (V_30_) was measured after 30 min. The FC and FS were subsequently calculated based on Equations (1) and (2).
(1)$$FC\left(\%\right)=\frac{V_{0}}{V}\times 100\%$$
(2)$$FS\left(\%\right)=\frac{V_{30}}{V_{0}}\times 100\%$$

### 2.9. Emulsifying Ability Measurements

Emulsification (EAI) and emulsification stability (ESI) were determined according to the method reported in Wang et al.’s study [29]. First, 4 mL of soybean oil was incorporated into 16 mL of protein solution (5 mg/mL), followed by homogenization at 10,000 rpm for 1 min to form an emulsion. When the emulsion was allowed to stand for 0 and 10 min, 50 μL of the emulsion was extracted from the lowermost 0.5 cm and introduced into a solution of SDS with a mass fraction of 0.1% in an amount of 5 mL. Following vigorous shaking and thorough mixing, absorbance at a wavelength of 500 nm was measured. The absorbance value of the mixture at 100 min was A0, and the absorbance value of the emulsion at 10 min was Aφ. A 0.1% SDS solution was used as a blank control. Emulsification (EAI) and emulsification stability (ESI) were calculated as follows (φ is the volume fraction of the oil phase in the emulsion, and N is the dilution ratio):(3)EAIm2g=2×2.303c×1−φ×104×A0×N,
(4)$$ESI\left(\%\right)=\frac{A_{\varphi }}{A_{0}}\times 100\%$$

### 2.10. Polarized Microscope Observations

The germinated black bean protein isolate solution was homogenized as described above, and the resulting homogenate sample was then transferred onto slides using a glass rod. The foaming microstructure was observed using a 10× eyepiece and a 40× objective lens using an XP-213 polarized biological microscope (NOVEL, Nanjing, China) [30].

### 2.11. In Vitro Digestibility Determination

The germinated black bean protein isolate was simulated using INFOGEST 2.0 and the method described in Jiang et al. [31], with slight modifications, for continuous in vitro gastrointestinal digestion. The proteins of the germinated black beans were extracted and prepared as a protein solution in deionized water with a concentration of 4% (*w*/*v*) for subsequent digestion. The protein sample (5 mL) was added to a solution of simulated saliva (5 mL, containing 150 U/mL α-salivary amylase) and 0.3 M CaCl_2_ (25 μL), followed by incubation for 2 min. The above digestion solution was supplemented with 10 mL of simulated gastric juice containing 4000 U/mL pepsin, followed by the addition of 5 μL of 0.3 M CaCl_2_ and a solution of HCl (5 mol/L) to adjust the pH to 2.5. The mixture was then subjected to digestion for a duration of 120 min. The mixed digestive solution then was supplemented with 20 mL of simulated intestinal fluid containing 10 mmol/L bovine bile salt and 200 U/mL trypsin, along with the addition of 40 μL of 0.3 M CaCl_2_. The pH value was adjusted to 7.0 using a solution of 5 mol/L NaOH, followed by digestion for a duration of 120 min. After digestion, enzymatic hydrolysis was terminated by immersing the samples in a boiling water bath for 10 min. Subsequently, the resulting solution was subjected to centrifugation at a speed of 8000× *g* per minute for 10 min. The supernatant was carefully collected and subsequently subjected to lyophilization prior to analysis.

The in vitro digestibility of the proteins was determined according to the method described in Wang et al. [32,33].

The concentration of L-leucine as a standard substance is linear (y = 0.0153x + 0.0778, R^2^ = 0.9997). The degree of digestion hydrolysis was calculated based on a standard curve of L-leucine.

### 2.12. Statistical Analysis

All tests were performed in triplicate and are expressed as mean ± SD deviation values. The data were analyzed using SPSS 27.0 (SPSS, Inc., Chicago, IL, USA). Analysis of variance (ANOVA) and Duncan’s test were used to compare data for the two groups. Three parallel experiments were conducted (*p* < 0.05). The figures were prepared using Origin 2022 (Northampton, MA, USA).

## 3. Results

### 3.1. Total Protein and Soluble Protein Contents of Germinated Black Beans

During germination, we monitored the germination rate of black beans, which reached over 90%, and the yield of isolated proteins extracted from ungerminated and germinated black beans was about 35% to 38.6%. Subsequently, we performed the determination of the total protein content and the soluble protein content of the germinated black beans. The total and soluble protein contents of the germinated black beans are shown in Figure 1. As the germination time increased, the germinating group showed higher concentrations of total protein and soluble protein compared to the non-germinating group. The total protein content of the germinated black beans reached its highest level at 24 h, with no further significant change up to G96. However, there were no statistically significant differences observed among the different time points (*p* > 0.05). The soluble protein content in germinated black beans exhibited a pattern of initially increasing and then decreasing. It reached its peak value at 72 h after sprouting, which aligns with the findings reported by Paucar-Menacho [34], De Souza-Rocha [35], and Concha [36]. The increase in protein content during seed germination may be attributed to the utilization of carbohydrates and fats for respiration, resulting in a reduction in dry mass while enhancing metabolic processes, thereby augmenting the relative protein content. Simultaneously, during the growth process, the synthesis of novel proteins also engenders an augmentation in protein content, thereby altering the protein profile of the seed [37,38,39,40]. The protein content is expected to decrease during germination, possibly because proteins are utilized to facilitate material metabolism through decomposition [41]. During seed germination, grain seeds undergo an enzymatic breakdown of their insoluble endosperm starch and non-soluble storage protein into soluble forms. This enzymatic breakdown facilitates the transport of nutrients to the embryo, meeting the nutritional demands for plant growth. Consequently, this process promotes seed germination and enhances seed resistance [42,43].

### 3.2. Amino Acid Composition and Content of the Germinated Black Bean Protein Isolate

Amino acids serve as the fundamental building blocks for protein synthesis in animal cells and actively participate in various physiological processes. These processes include nutrient metabolism, glucose and lipid metabolism, hormone regulation, and signal transduction. Additionally, amino acids play crucial roles in obesity, diabetes, cardiovascular diseases, and cerebrovascular diseases [44,45,46]. The amino acid composition of the germinated black bean protein isolate exhibited significant alterations following the sprouting treatment, as evidenced by the data presented in Table 1. The total amino acid content of the germinated black bean protein isolate significantly increased (*p* < 0.05) compared to the ungerminated black bean protein isolate. It reached a maximum value of 98.072 g/100 g after 96 h of sprouting, which was 1.16 times higher than that of its ungerminated counterpart. The nutritional value of protein can be evaluated based on the content and composition of essential amino acids. In the case of the germinated black bean protein isolate, an increase in the germination time resulted in a significant increase (*p* < 0.05) in the total content of essential amino acids. A maximum value of 31.893 g/100 g was observed after 96 h of germination, which was 1.14 times higher than the content of the ungerminated black bean protein isolate. There are some relationships between protein changes and the total amino acid. Endogenous enzymatic hydrolysis might present dominant effects on protein release (Figure 1). In turn, part of their hydrolysates, such as peptides or amino acids, will be present in black beans during their metabolism or synthesis of other products. The significant increment of total amino acids (*p* < 0.05) (Table 1) and protein release (Figure 1) will be a worthy topic of discussion in future. These findings align with Zhang’s previous research [47]. As shown in the table, the black bean protein isolate contains significant amounts of hydrophobic and acidic amino acids, including Ala, Val, Ile, Leu, Pro, Phe, and Met. Previous studies indicated that these hydrophobic amino acids possess high numbers of hydrogen atoms which can efficiently scavenge free radicals and protect against oxidative damage [48,49].

The amino acid Met, which contains sulfur and exhibits a resonance structure, demonstrates the ability to eliminate free radicals through single-electron transfer and maintain oxidative stability [50]. Acidic amino acids, such as Asp and Glu, play a significant role in maintaining metabolic homeostasis. Glu has the ability to enhance liver glycogen synthesis and promote skeletal muscle protein homeostasis. On the other hand, Asp is involved in regulating hormone levels, thus influencing the functioning of the nervous system [51,52]. In addition, acidic amino acids have the ability to eliminate free radicals by transferring hydrogen atoms, thereby disrupting the oxidation chain reaction. They also have the ability to chelate metal ions and weaken the oxidation of these ions, thereby preventing the production of peroxide [53,54]. With an increase in the germination time, the levels of hydrophobic and acidic amino acids in the germinated black bean protein isolate were found to increase significantly compared to those in the ungerminated black bean protein isolate (*p* < 0.05). This suggests that the germinated black bean protein isolate may possess enhanced antioxidant and metabolic regulatory functions compared to its ungerminated counterpart. To enhance the nutritional value of amino acids and proteins, it is recommended to consume a germinated black bean protein isolate along with other animal foods. This combination can help increase the overall nutritional value of the diet [55].

### 3.3. Effect of Different Germination Times on Isolated Protein Structure

#### 3.3.1. SDS-PAGE Analysis

The molecular weight of the germinated black bean protein isolate was determined using SDS-PAGE. The molecular weight of the germinated black bean protein isolate samples ranged from 18.4 kDa to 116 kDa and primarily consisted of globulin, specifically 7S globulin and 11S globulin. Based on Teraishi’s findings and considering the relative molecular weight, the black bean protein isolate’s 7S globulin group was classified into α′, α, and β subunits. Additionally, the 11S globulin component was further categorized into acidic and basic subunits [56]. It can be seen from our results in Figure 2 that the electrophoretic band density of the germinated black bean protein isolate samples changes such that the number of protein bands with a larger molecular weight decreases or density decreases while that of protein bands with a smaller molecular weight appear or density increases. An electrophoretic map reveals that the strip densities of the α′, α, β, and acid subunits noticeably decrease after germination. This change becomes more pronounced as the germination time increases.

The content ratio of 11S to 7S components directly affects the quality and nutritional value of black bean protein due to their different amino acid compositions. An optical density analysis of the germinated black bean protein isolate samples was conducted using Image J software and an SDS-PAGE electrophoretic map. The results in Table 2 reveal that as the germination time increased, the relative content of 7S globulin decreased gradually. Additionally, the ratio of the 11S (acid + basic) component to the 7S (α′ + α + β) component showed a gradual increase. According to previous studies, 11S globulin has been found to be rich in sulfur-containing amino acids, with a content that is five to six times higher than that of 7S globulin. This indicates that 11S globulin has higher protein quality and can provide greater nutritional value [57].

#### 3.3.2. Spectral Analysis

The FTIR results for the germinated black bean protein isolate are presented in Figure 3a. The absorption peak wave number of the germinated black bean protein isolate in the infrared spectrum did not exhibit a noticeable blue or red shift, and no new absorption peak was observed. This suggests that the chemical bond composition of the black bean protein molecules remained unchanged after the sprouting treatment, and no new chemical substances were generated. Therefore, this can be attributed to physical denaturation. However, the absorption peak intensity of the isolated protein from germinated black beans was observed to be different. In comparison to the non-germinated group, the absorption peak intensity of the germinated black bean group was found to be stronger. This figure also shows characteristic peaks in the amide I band (1655 cm^−1^) and amide III band (1234 cm^−1^) of the black bean protein isolate. These peaks are generated by a N-H stretching vibration, a C-N stretching vibration, and a N-H bending vibration [58,59]. The protein’s secondary structure is supported by hydrogen bonds between peptide bonds. The two most common secondary structures are the α-helix and β-fold [60,61]. Table 3 shows the results obtained using a Fourier deconvolution spectral fitting analysis of the amide I band. The region associated with the secondary structure of the amide I band includes the α-helix (1646~1664 cm^−1^), random coil (1637~1645 cm^−1^), β-sheet (1615~1637 cm^−1^, 1682~1700 cm^−1^), and β-turn (1664~1681 cm^−1^) [62]. According to Table 3, the primary secondary structure of the black bean protein isolate is β-sheets (35.91%). Over time, there was a gradual decrease in the levels of β-sheet and β-turn in the amide I band of the germinated black bean protein isolate, while the α-helix content increased. This suggests that the structure of the germinated black bean protein underwent changes, possibly due to the disruption of hydrogen bonds between protein peptide chains and the alteration of the protein isolate’s secondary structure.

Proteins exhibit a distinct absorption spectrum in the ultraviolet region, with the absorption peak (260–280 nm) corresponding to the absorption of aromatic amino acid residues [63]. As depicted in Figure 3b, UV spectrum scanning of the germinated black bean protein isolate revealed an absorption peak at 260–280 nm. Moreover, the UV absorption intensity of the germinated black bean protein isolate exhibited a continuous increase, suggesting that the protein structure was stretched.

Fluorescence spectra can indicate changes in protein conformation, which is primarily influenced by the environmental polarity of aromatic amino acids, particularly tryptophan. According to Figure 3c, after 72 h of germination, the maximum emission wavelength of the germinated black bean protein isolate shifted in the red direction (from 235 nm to 330 nm). This shift suggests that the tryptophan residues in the sample were in a polar environment. The reason for this change may be that germination changes the spatial conformation of the black bean protein isolate, increases the polar environment of aromatic amino acid residues, and extends the structure of the peptide chain so that the tertiary structure of the germinated black bean protein isolate stretches more. During germination, the tertiary structure of the germinated black bean protein isolates unfolds, causing the peptide chain structure to stretch. As a result, residues that were initially buried within the hydrophobic environment inside the molecule are gradually exposed to the external hydrophilic environment. This exposure leads to changes in fluorescence intensity and shifts in the maximum excitation wavelength. The decrease in the fluorescence intensity of the germinated black bean protein isolate can be attributed to alterations in the spatial conformation of the protein structure and the microenvironment surrounding the aromatic amino acids, leading to fluorescence quenching [64].

#### 3.3.3. Particle Size and ζ-Potential

According to Figure 4, the germinated black bean protein isolate exhibits a smaller particle size compared to its non-germinated counterpart. Moreover, the germinated black bean protein isolate solution demonstrates a more uniform dispersion of protein particles. A possible explanation for this lies in the fact that as the germination time increases, a fraction of the germinated black bean protein isolate undergoes hydrolysis by endogenous enzymes, resulting in the formation of smaller components. Consequently, this enzymatic hydrolysis weakens the interaction between protein and water molecules, leading to a more homogeneous dispersion in the solution [65]. The electrical properties and potential values of proteins are intricately linked to the amino acid composition and the stability of protein solution micelles.

The side chains of the protein molecules consist of both polar and non-polar groups. The hydrophilic polar groups are exposed on the surface of the protein molecules, resulting in the protein surface being charged. The charge carried by amino acids on the protein surface, as well as their positive and negative properties, influences the surface potential of the protein solution. A positive potential value of a protein solution is observed when the surface of the protein molecules contains a higher proportion of positively charged amino acids compared to negatively charged ones; conversely, when there are more negatively charged amino acids on the surface of protein molecules, the protein solution shows electronegativity [66]. In general, the majority of protein molecules exhibit electronegativity in a neutral environment [67].

The ζ-potentials of the germinated black bean protein, as presented in Table 4, exhibit a negative charge, thereby indicating an abundance of negatively charged sites on the surface of the black bean protein isolate. With the prolonged germination time, there was an increase in the absolute value of the ζ-potential. A significant difference in ζ-potential between the germinated black bean protein isolate and the ungerminated black bean protein isolate was observed at 72 h (*p* < 0.05). The findings reveal that an extended germination period led to an augmented exposure of charged amino acid residues on the protein structure’s surface, consequently resulting in an amplified distribution of charge across the surface [68]. A small absolute value of the ζ-potential in the protein solution indicates a lower number of homogeneous charges on the surface of the protein molecules, resulting in reduced solution stability due to decreased electrostatic repulsion forces and an increased tendency for protein aggregation [69]. Conversely, an increase in the absolute value of the ζ-potential of a protein molecular solution signifies an augmentation in the same charges on the surface of the protein molecules, leading to enhanced mutual repulsion forces between these charges and consequently reducing the intermolecular aggregation forces among proteins. The ζ-potential of the germinated black bean protein isolate exhibited an increase, suggesting enhanced stability in the properties of the solution.

### 3.4. Solubility and Turbidity of the Germinated Black Bean Protein Isolate

The solubility of proteins is a crucial manifestation of their hydration and serves as an excellent indicator for assessing their application properties, thereby influencing various functional characteristics. The solubility of the germinated black bean protein isolate samples increased to varying degrees with an increase in the germination time at a pH of 7, as depicted in Figure 5a. At 72 h of germination, the solubility of the germinated black bean protein isolate reached its peak, exhibiting a significant difference compared to other groups (*p* < 0.05). The increased solubility of the germinated black bean protein isolate may be attributed to several factors: the enzymatic hydrolysis of proteins during sprouting, the partial unfolding of protein molecules, and enhanced ionic interactions, all contributing to improved protein solubility [37]. The solubility characteristics of proteins isolated from germinated black beans vary under different pH conditions, with the lowest solubility observed at pH values of 4–5. This is attributed to the predominant presence of 7S and 11S globulins in the separation protein of black soybean. In proximity to their isoelectric point, the protein molecules exhibit a near-zero net charge, resulting in diminished electrostatic repulsion between them. Consequently, this leads to protein aggregation and precipitation with minimal solubility, following a U-shaped curve. Relevant research findings were also documented by Yang et al. [70].

Turbidity serves as an indicator of the extent to which particles in a solution obstruct the passage of light while also providing insights into the dispersion and aggregation states of these particles. Turbidity is quantified by absorbance values, with higher absorbance values indicating greater turbidity [71]. Research findings indicated that the solubility of smaller protein molecules was enhanced due to the increased surface area available for interaction with water molecules [72]. In addition, a decrease in the molecular weight of proteins also increases the adsorption rate of proteins to oil–water interfaces [73]. According to Figure 5b, the turbidity of the isolated proteins from germinated black beans gradually decreased with an increasing germination time. The protein isolated after 72 h of germination forms smaller aggregates in the aqueous phase and represents a more transparent solution.

### 3.5. Emulsification and Foamability

Emulsification refers to the ability of hydrophobic parts to orient toward lipids and the ability of polar parts to orient toward the aqueous phase in an oil–water mixture. Good emulsification and emulsion stability can extend the shelf life of food. The emulsifying property and emulsion stability of the germinated black bean protein exhibited an increasing trend with prolonged germination time, as illustrated in Figure 6a. The emulsifying property of the germinated black bean protein reaches its maximum value at 24 h of germination, exhibiting no significant variation during the subsequent stages of germination (*p* < 0.05). During the germination process, degradation of the storage proteins in black beans occurs, leading to the unfolding of the protein structure and the exposure of internal hydrophobic groups. This phenomenon significantly enhances diffusion and adsorption capabilities at oil–water interfaces, thereby resulting in heightened emulsifying properties and improved emulsion stability [74].

The foaming ability and foam stability, in addition to the aforementioned factors, are also related to an increase in the solubility of the protein. These properties have the potential to improve the softness and texture of foods [75]. Figure 6b shows that the germination treatment had a significant effect on the foaming ability and foam stability of the black bean isolated proteins. As the germination time increased, the foaming ability of the isolated proteins increased and reached a maximum (52.44%) at 96 h after germination (*p* < 0.05). This may have occurred for the following reasons: during the black beans’ germination period, some protein molecules in the black beans were hydrolyzed to form a low-molecular-weight subunit structure, the solubility of the protein isolate was increased, and the adsorption capacity of the air–water interface was strengthened, thereby improving the foaming properties of the germinated black bean protein isolate.

### 3.6. Polarized Microscope Observations

The microstructure of the emulsion droplets formed by the germinated black bean protein is shown in Figure 7A. There are significant differences between the microstructure of the emulsion formed by the germinated black bean protein and that of the non-germinated black bean protein. The emulsion droplets formed by the separation of separation from the non-germinated black beans are irregular and exhibit significant flocculation and stratification. As the germination time increases, the size of the emulsion droplets decreases, and they become evenly distributed and have a regular spherical shape. After allowing the emulsion to stand for 30 min, the coagulation and stratification phenomena of the protein produced from non-germinated black beans increased. However, the emulsion formed from the germinated black bean protein maintained a stable and uniform droplet distribution.

The microstructure of the homogenized germinated black bean protein solution is shown in Figure 7B. As the germination time increases, the foam of the homogenized solution becomes relatively uniform, with a thicker liquid film which partially increases the stability of the foam. After standing for 30 min, the separated protein foam structure of the germinated black beans remains in good condition. The possible reason for this was that the structure of the separated protein from the germinated black beans changed, increasing its adsorption ability at the interface and suppressing the speed of bubble discharge, resulting in a more stable foam structure [76].

### 3.7. In Vitro Digestibility

The OPA method is often used to determine the concentration of free amino acids in proteins to characterize the extent of protein hydrolysis during digestion. A higher concentration of free amino acids indicates a higher degree of protein digestion and hydrolysis [32]. From Figure 8, it can be seen that the gastric digestion rate of the proteins isolated from germinated black beans shows a decreasing trend, followed by an increasing trend with an increasing germination time. At 72 h after germination, the gastric digestion rate of the proteins isolated from germinated black beans reached its maximum value (12.85 ± 0.44%). During the intestinal digestion phase, the in vitro digestibility of the germinated black bean protein increased with an increasing germination time. It reached its maximum digestion rate after 72 h (41.50 ± 0.97%), a rate 1.35 times higher than that of the non-germinated black bean protein (30.74 ± 1.93%). The increased in vitro digestibility of black bean isolated protein after germination may be due to the fact that germination can activate endogenous protease activity and also lead to increased protein solubility [77]. The in vitro digestibility of proteins was an important feature to characterize the protein quality, and proteins with higher in vitro digestibility are considered of high quality, because proteolysis helps the release of amino acids from the protein backbone, meaning that the protein can be better digested and absorbed by the body [78]. This fully proves that the protein isolated from germinated black beans is hydrolyzed more thoroughly than the protein from non-germinated black beans, making it more suitable for absorption and utilization by the body. This result is consistent with the findings of Bautista-Exposito et al. [79] and Devi et al. [80].

## 4. Conclusions

This study investigates the effects of germination on the structure, amino acid composition, and functional properties of proteins isolated from black soybean. The findings indicate that germination resulted in an increase in both the total and soluble protein contents of black beans, while SDS-PAGE demonstrated an increase in the proportion of 11S and 7S globulin subunits. After germination, the particle size of black bean protein isolate decreased in solution, while the absolute value of zeta potential increased. The above results show that the stability of the solution was improved. The contents of β-sheet and β-turn gradually decreased, while the content of α-helix increased, and the fluorescence spectrum of black bean protein isolate showed a red shift phenomenon, indicating that the structure of the protein isolate and its polypeptide chain were prolonged, and the foaming property, emulsification property and in vitro digestibility were significantly improved after germination. Therefore, germination not only improves functional properties, but also nutritional content. The above results fully demonstrate the positive improvement effect of germination on the separation of protein from black beans, making them more suitable for processing and consumption. Based on the above results, this area represents a good research direction for the separation and purification of bioactive peptides from the products of germinated black bean protein isolate after digestion in vitro. This can serve as a reference for the future processing and utilization of separated black bean protein.

## Figures and Tables

**Figure 1 foods-13-00488-f001:**
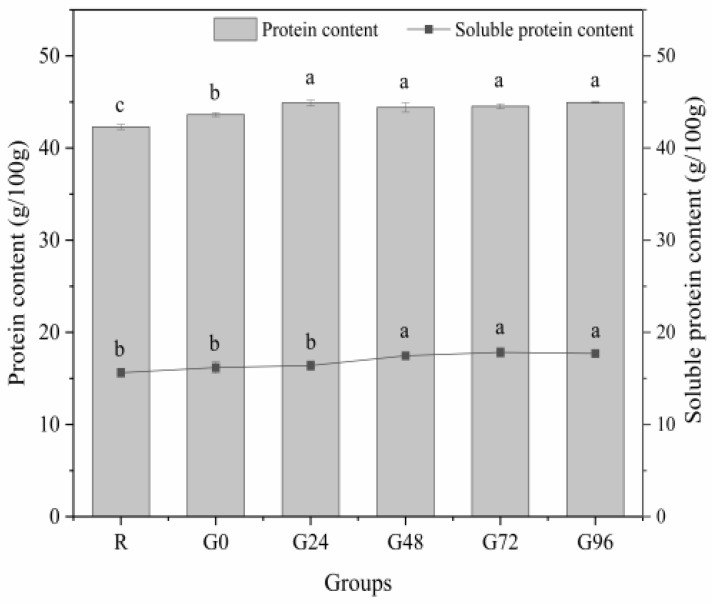
Total protein and soluble protein content of the germinated black bean powder. Different letters (a–c) indicate significant differences between the mean values of different samples (*p* < 0.05).

**Figure 2 foods-13-00488-f002:**
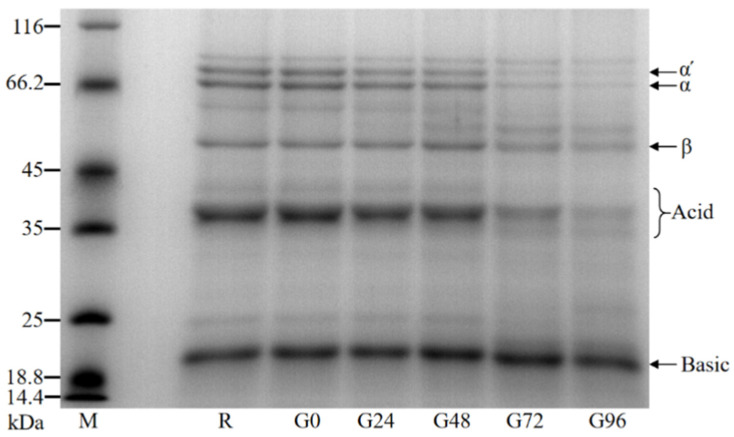
SDS-PAGE electrophoresis diagram of the germinated black bean isolated protein.

**Figure 3 foods-13-00488-f003:**
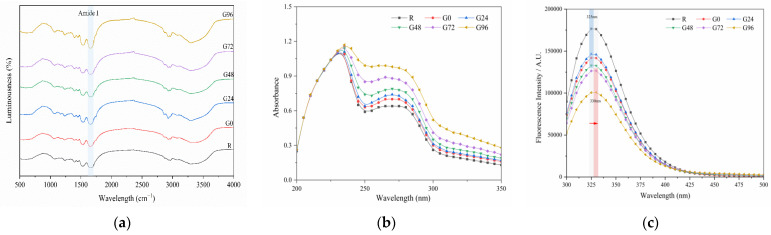
Fourier-transform infrared spectroscopy (**a**); ultraviolet spectrum (**b**) and fluorescence spectrum (**c**) of the germinated black bean isolated protein. Note: The red arrow in Figure c indicates a redshift (from 235 nm to 330 nm).

**Figure 4 foods-13-00488-f004:**
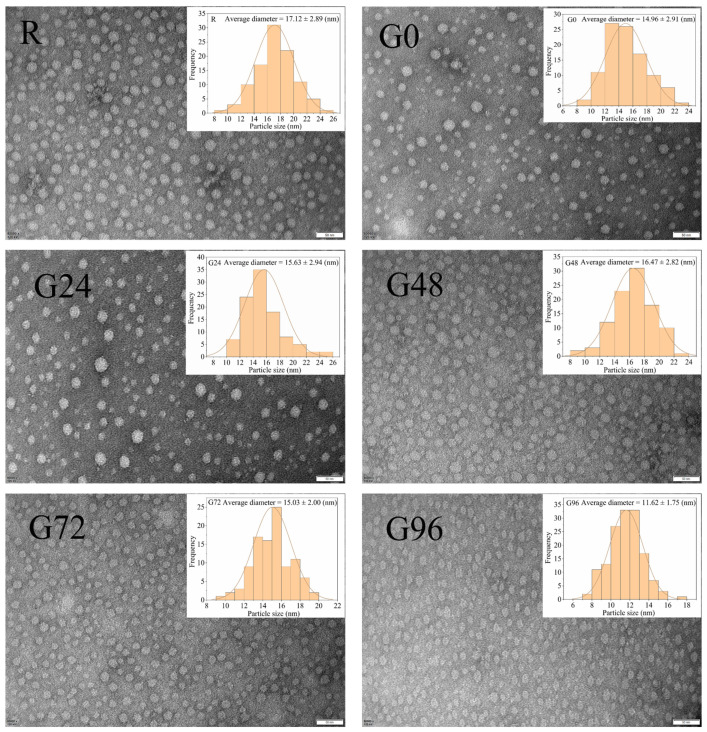
TEM of proteins isolated from germinated black bean.

**Figure 5 foods-13-00488-f005:**
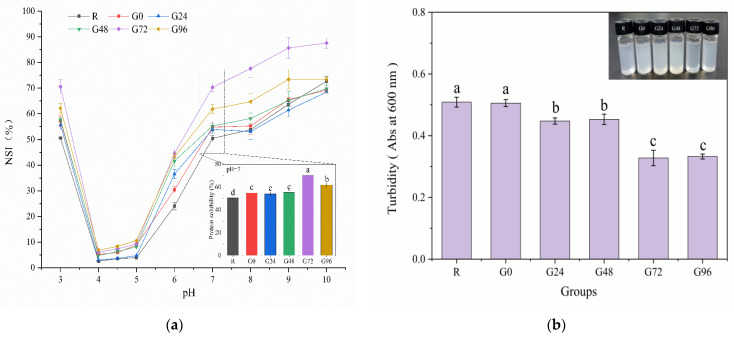
Solubility (**a**) and turbidity (**b**) of the germinated black bean protein isolate. Note: Different letters (a–c) indicate significant differences between the mean values of different samples (*p* < 0.05).

**Figure 6 foods-13-00488-f006:**
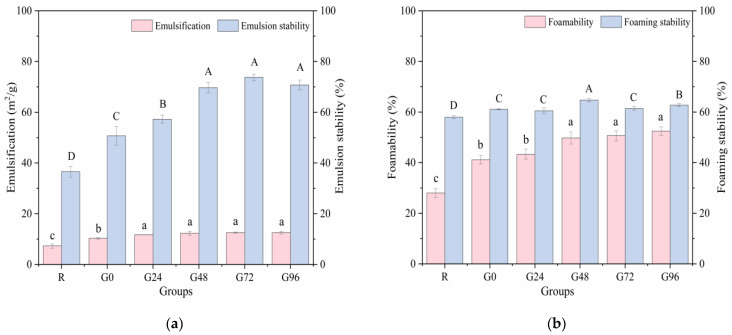
Emulsification (**a**) and foamability (**b**) of the germinated black bean protein isolate. Note: Different letters indicate significant differences between the mean values of different samples (*p* < 0.05).

**Figure 7 foods-13-00488-f007:**
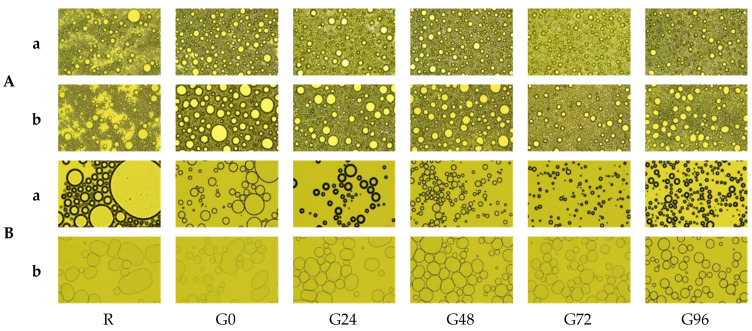
Emulsification state (**A**) and foaming changes (**B**) of the germinated black bean protein isolate’s microstructure ((**a**) Picture taken after 1 min of homogenization; (**b**) picture taken after 1 min of homogenization and standing for 30 min; the magnification is 40 times).

**Figure 8 foods-13-00488-f008:**
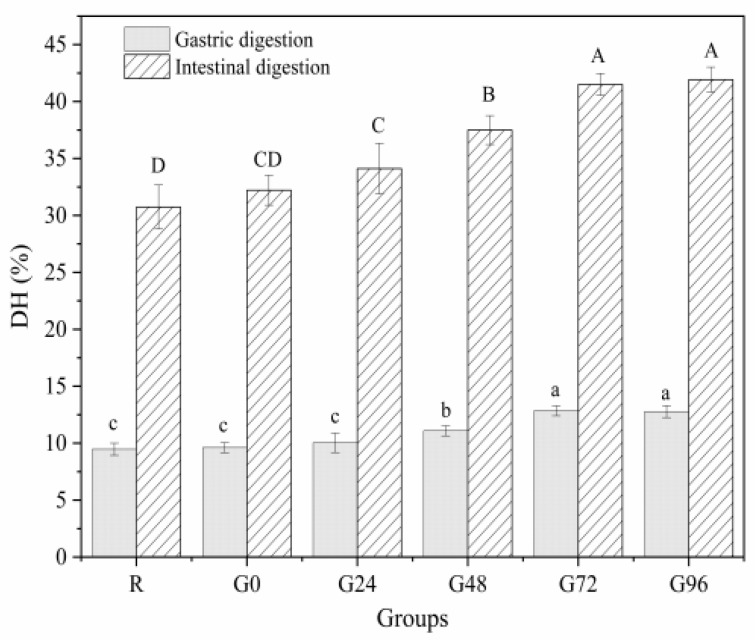
In vitro digestibility of germinated black bean protein isolate. Note: Different letters indicate significant difference between the mean values of different samples (*p* < 0.05).

**Table 1 foods-13-00488-t001:** Amino acid composition and content of proteins isolated from germinated black beans.

Amino Acid Species		Amino Acid Content (g/100 g)					
		R	G0	G24	G48	G72	G96
Hydrophobic Amino Acid	Ala	3.793 ± 0.015 ^e^	3.775 ± 0.011 ^e^	3.898 ± 0.014 ^d^	4.072 ± 0.015 ^c^	4.267 ± 0.009 ^a^	4.105 ± 0.014 ^b^
	* Val	4.421 ± 0.008 ^e^	4.369 ± 0.010 ^f^	4.586 ± 0.022 ^d^	4.703 ± 0.012 ^b^	4.908 ± 0.017 ^a^	4.663 ± 0.025 ^c^
	* Ile	1.152 ± 0.019 ^de^	1.138 ± 0.013 ^e^	1.370 ± 0.021 ^a^	1.312 ± 0.005 ^b^	1.246 ± 0.007 ^c^	1.174 ± 0.016 ^d^
	* Leu	4.024 ± 0.016 ^f^	4.078 ± 0.023 ^e^	4.189 ± 0.010 ^d^	4.287 ± 0.009 ^c^	4.534 ± 0.016 ^a^	4.334 ± 0.021 ^b^
	Pro	7.284 ± 0.024 ^f^	7.325 ± 0.006 ^e^	7.541 ± 0.011 ^d^	7.738 ± 0.017 ^c^	8.211 ± 0.007 ^a^	7.874 ± 0.011 ^b^
	* Phe	4.091 ± 0.014 ^f^	4.127 ± 0.012 ^e^	4.294 ± 0.013 ^d^	4.441 ± 0.022 ^c^	4.659 ± 0.018 ^a^	4.571 ± 0.008 ^b^
	* Met	3.836 ± 0.021 ^e^	4.361 ± 0.010 ^d^	4.393 ± 0.018 ^c^	4.431 ± 0.010 ^b^	4.400 ± 0.005 ^c^	4.659 ± 0.020 ^a^
	Total Hydrophobic Amino Acids	28.601 ± 0.012 ^f^	29.173 ± 0.025 ^e^	30.271 ± 0.014 ^d^	30.984 ± 0.015 ^c^	32.225 ± 0.016 ^a^	31.380 ± 0.020 ^b^
Polar Amino Acid	* Thr	3.511 ± 0.007 ^f^	3.584 ± 0.011 ^e^	3.663 ± 0.014 ^d^	3.800 ± 0.015 ^c^	3.943 ± 0.006 ^a^	3.871 ± 0.011 ^b^
	Ser	4.767 ± 0.016 ^f^	5.160 ± 0.012 ^e^	5.213 ± 0.018 ^d^	5.367 ± 0.019 ^c^	5.498 ± 0.005 ^b^	5.536 ± 0.008 ^a^
	Gly	3.547 ± 0.012 ^f^	3.856 ± 0.013 ^e^	3.909 ± 0.009 ^d^	3.987 ± 0.012 ^c^	4.072 ± 0.009 ^b^	4.118 ± 0.007 ^a^
	Tyr	3.504 ± 0.009 ^e^	3.440 ± 0.012 ^f^	3.550 ± 0.011 ^d^	3.653 ± 0.011 ^c^	3.908 ± 0.002 ^a^	3.709 ± 0.010 ^b^
	Total Polar Amino Acid	15.329 ± 0.101 ^f^	16.040 ± 0.020 ^e^	16.335 ± 0.017 ^d^	16.807 ± 0.012 ^c^	17.421 ± 0.011 ^a^	17.234 ± 0.011 ^b^
Acidic Amino Acid	Asp	11.760 ± 0.008 ^d^	11.449 ± 0.018 ^f^	11.620 ± 0.013 ^e^	12.009 ± 0.014 ^c^	12.684 ± 0.011 ^a^	12.258 ± 0.020 ^b^
	Glu	15.925 ± 0.013 ^e^	19.816 ± 0.007 ^c^	19.896 ± 0.015 ^b^	19.821 ± 0.005 ^c^	19.323 ± 0.007 ^d^	21.136 ± 0.015 ^a^
	Total Acidic Amino Acid	27.685 ± 0.009 ^f^	31.265 ± 0.014 ^e^	31.516 ± 0.012 ^d^	31.830 ± 0.009 ^c^	32.007 ± 0.009 ^b^	33.394 ± 0.019 ^a^
Basic Amino Acid	* Lys	4.956 ± 0.011 ^e^	5.712 ± 0.011 ^d^	5.739 ± 0.012 ^c^	5.760 ± 0.002 ^b^	5.743 ± 0.011 ^bc^	6.123 ± 0.009 ^a^
	* His	2.063 ± 0.011 ^d^	2.336 ± 0.008 ^b^	2.277 ± 0.012 ^c^	2.355 ± 0.009 ^b^	2.286 ± 0.013 ^c^	2.498 ± 0.012 ^a^
	Arg	5.604 ± 0.010 ^e^	7.030 ± 0.009 ^b^	6.937 ± 0.015 ^c^	6.950 ± 0.009 ^c^	6.604 ± 0.006 ^d^	7.398 ± 0.011 ^a^
	Total Basic Amino Acid	12.623 ± 0.012 ^e^	15.078 ± 0.011 ^b^	14.953 ± 0.012 ^c^	15.065 ± 0.021 ^b^	14.633 ± 0.018 ^d^	16.019 ± 0.015 ^a^
EAA		28.054 ± 0.011 ^f^	29.705 ± 0.008 ^e^	30.511 ± 0.010 ^d^	31.089 ± 0.017 ^c^	31.719 ± 0.011 ^b^	31.893 ± 0.014 ^a^
TAA		84.238 ± 0.019 ^f^	91.556 ± 0.015 ^e^	93.075 ± 0.011 ^d^	94.686 ± 0.012 ^c^	96.25 ± 0.009 ^b^	98.027 ± 0.012 ^a^

Note: Total essential amino acids, EAA; total amino acids, TAA; essential amino acids are marked with “*”. Different letters (^a^–^f^) indicate significant differences between the mean values of different samples (*p* < 0.05).

**Table 2 foods-13-00488-t002:** The contents of 7S and 11S globulin of different soybean.

Protein	Subunit	Relative Content of Isolated Protein Subunits at Different Germination Times/%					
		R	G0	G24	G48	G72	G96
7S Globulin	a′	3.69	4.00	2.93	2.71	2.24	2.11
	a	3.75	3.92	3.68	3.41	2.66	2.44
	β	3.37	3.46	3.64	3.79	3.50	3.43
	a′ + a + β	10.80	11.37	10.24	9.90	8.40	7.98
11S Globulin	Acid	19.83	20.63	20.04	19.39	17.48	16.50
	Basic	10.69	11.21	10.87	11.19	12.77	12.83
	Acid + Basic	30.52	31.84	30.91	30.58	30.25	29.33
7S + 11S		41.32	43.22	41.15	40.48	38.65	37.31
11S/7S		2.82	2.80	3.02	3.09	3.60	3.67

**Table 3 foods-13-00488-t003:** Secondary structure content of isolated proteins from germinated black bean.

	β-Sheet (%)	Random Coil (%)	α-Helix (%)	β-Turn (%)
R	35.91 ± 0.27 ^a^	17.22 ± 0.13 ^a^	15.81 ± 0.62 ^c^	24.87 ± 0.18 ^ab^
G0	36.21 ± 0.79 ^a^	15.98 ± 2.03 ^a^	15.67 ± 0.86 ^c^	26.12 ± 2.35 ^a^
G24	33.05 ± 2.69 ^b^	15.66 ± 1.80 ^a^	23.45 ± 1.43 ^b^	25.53 ± 2.48 ^ab^
G48	30.87 ± 1.72 ^bc^	15.42 ± 1.54 ^a^	27.01 ± 2.08 ^a^	22.47 ± 1.48 ^bc^
G72	29.79 ± 0.68 ^c^	16.59 ± 0.28 ^a^	29.05 ± 0.63 ^a^	19.94 ± 0.51 ^c^
G96	28.99 ± 0.69 ^c^	17.33 ± 2.32 ^a^	28.94 ± 1.08 ^a^	20.17 ± 1.85 ^c^

Note: Different letters (^a^–^c^) indicate significant difference between the mean values of different samples (*p* < 0.05).

**Table 4 foods-13-00488-t004:** The ζ-potentials of the germinated black bean protein.

Groups	R	G0	G24	G48	G72	G96
Particle size (nm)	17.12 ± 2.89	14.96 ± 2.91	15.63 ± 2.94	16.47 ± 2.82	15.03 ± 2.00	11.62 ± 1.75
ζ-potential (mV)	−22.33 ± 0.57 ^b^	−23.23 ± 1.39 ^ab^	−22.97 ± 0.90 ^ab^	−23.17 ± 0.40 ^ab^	−24.47 ± 0.65 ^a^	−24.13 ± 0.75 ^a^

Note: Different letters (^a^, ^b^ indicate significant differences between the mean values of different samples (*p* < 0.05).

## Data Availability

The original contributions presented in the study are included in the article, further inquiries can be directed to the corresponding author.

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
