# Peer review of "Effects of Germination on the Structure, Functional Properties, and In Vitro Digestibility of a Black Bean (Glycine max (L.) Merr.) Protein Isolate"

_foods, 2024, doi:10.3390/foods13030488_

Round 1

Reviewer 1 Report

Comments and Suggestions for Authors

Pag. 1, line 31

Germination not only improves functional properties, but also nutritional content

Pag. 4. Line 115-125

Line 113: Have the authors determined how much of the nitrogen determined by Kjeldahl  method  is protein?

I think it is more interesting to describe the determination of soluble protein content, rather than total protein, which is clearly described in the AOAC methods. In general, sprouts have an appreciable fat content, which interferes with the determination of other analytes. The authors do not specify whether the sample was previously degreased.

Pag 6, line 223

“After digestion, enzymatic hydrolysis was terminated by immersing 223 the samples in a boiling water bath for 10 minutes”

What you actually determine is the degree of hydrolysis

Pg. 7, Figure 1.

Figure 1 shows that the total protein contents of the sprouts are high (around 40%), however the authors do not mention the yield in sprouts and the yield in obtaining the protein isolate from the sprouts. I believe that research should be carried out with a practical application approach.

Pg. 7, lines 244-246

Figure 1, shows that the total protein content increases up to 24 hours of germination, subsequently this content is maintained, it does NOT decrease, for this reason there is no significant difference. It is suggested to review the expression of these results (lines 244-246)

Pg 8, table 1

It is suggested that the authors clarify or better explain the increase in amino acid content with germination time, despite the fact that the total protein is maintained after 24 h of germination. Considering that total protein is the result of the total of  amino acids.

Pag 9, line 330

It is suggested to improve the wording of this paragraph: "Conversely, the relative content of the basic subunit of 11S globulin increased", since the aforementioned increase is only observed up to 24 h of germination

Pag 13, line 477

The foaming ability and foams stability, in addition to the aforementioned factors, is also related to the increase in the solubility of the protein.

Pg 13, line 481.

Figure 6 shows that the maximum foaming capacity is reached at 72 hours NOT at 96 hours. – review

Pag 16. Fig 8

Using exogenous enzymes (papain and flavourzyme) a DH of 35% is achieved. In this study, only the germination process reached a DH of 40%. To verify the goodness of germination in the digestibility of the protein, perhaps a comparison standard such as casein should have been used.

Author Response

Dear Reviewer :

We feel great thanks for your professional review work on our article. As you are concerned, there are several problems that need to be addressed. According to your nice suggestions, we have made extensive corrections to our previous draft, and the specific concerns have been numbered. Our response is given in normal font and changes to the manuscript are given in the blue text.

Q1: Pag. 1, line 31 

Germination not only improves functional properties, but also nutritional content

A: We sincerely appreciate the valuable comments. We carefully examined the statements and modified the incorrect statements.

Q2: Pag. 4. Line 115-125

Line 113: Have the authors determined how much of the nitrogen determined by Kjeldahl  method  is protein?

I think it is more interesting to describe the determination of soluble protein content, rather than total protein, which is clearly described in the AOAC methods. In general, sprouts have an appreciable fat content, which interferes with the determination of other analytes. The authors do not specify whether the sample was previously degreased.

A: We sincerely appreciate this valuable comment. Thank you for your reminder. In our resubmitted manuscript, this part has been modified.

Q3: Pg. 7, Figure 1.

Figure 1 shows that the total protein contents of the sprouts are high (around 40%), however the authors do not mention the yield in sprouts and the yield in obtaining the protein isolate from the sprouts. I believe that research should be carried out with a practical application approach.

A: We think this is an excellent suggestion. In the experiment, we measured the relevant indicators, the yield of sprouts was about 90%, and the yield of germinated black bean isolated protein was about 35% -38.6%.

Q4: Pg. 7, lines 244-246

Figure 1, shows that the total protein content increases up to 24 hours of germination, subsequently this content is maintained, it does NOT decrease, for this reason there is no significant difference. It is suggested to review the expression of these results (lines 244-246)

Pg 8, table 1

It is suggested that the authors clarify or better explain the increase in amino acid content with germination time, despite the fact that the total protein is maintained after 24 h of germination. Considering that total protein is the result of the total of  amino acids.

A: Authors accepted your suggestive comments. The total protein content of the sprouted black beans indicated maximum at 24 h (p < 0.05) and then decreased with a no significant difference (p > 0.05) compared to the ungerminated group (Figure 1). After germination at a specific time, endogenous enzymatic hydrolysis will present dominant effects on protein and more soluble protein might be released (Figure 1), in turn, part of their soluble proteins or their deep hydrolysates (namely kinds of amino acid or peptides) will involve in life metabolism or synthesis for other products. Therefore there are significant increment of total amino acids (p < 0.05) (Table 1). The specific germination time and exact mechanism for the balance between degree of the total protein hydrolysis and increment of total amino acid are a valuable topic to discuss in future. And particular statements was addressed in the paper.

Q5: Pag 9, line 330

It is suggested to improve the wording of this paragraph: "Conversely, the relative content of the basic subunit of 11S globulin increased", since the aforementioned increase is only observed up to 24 h of germination

A: We sincerely appreciate the valuable comments, the statement has been deleted.

Q6: Pag 13, line 477

The foaming ability and foams stability, in addition to the aforementioned factors, is also related to the increase in the solubility of the protein.

A: We sincerely appreciate the valuable comments. We carefully examined the statements and modified the incorrect statements.

Q7: Pg 13, line 481.

Figure 6 shows that the maximum foaming capacity is reached at 72 hours NOT at 96 hours. – review

A: Thanks for your suggestion. We examined the data in the figure, that the maximum foamability was reached at 96 hours, and the maximum foaming stability was reached at 48 hours.

Q8: Pag 16. Fig 8

Using exogenous enzymes (papain and flavourzyme) a DH of 35% is achieved. In this study, only the germination process reached a DH of 40%. To verify the goodness of germination in the digestibility of the protein, perhaps a comparison standard such as casein should have been used.

A: We think this is an excellent suggestion. This part of the study refers to wang's method (32). The OPA method is often used to determine the concentration of free amino acids in proteins to characterize the extent of protein hydrolysis during digestion. A higher concentration of free amino acids indicates a higher degree of protein digestion and hydrolysis.

We tried our best to improve the manuscript and made some changes marked in blue in revised paper which will not influence the content and framework of the paper. We appreciate for Reviewers’ warm work earnestly, and hope the correction will meet with approval. Once again, thank you very much for your comments and suggestions.

Reviewer 2 Report

Comments and Suggestions for Authors

The manuscript present very intersting findings regarding germination importance of black beans for their nutritional and techno-functional properties. The manuscript is overall well written and presented and will have a significant contribution to the field. Some points should be addressed, however, before publishing. 

Please mention the latin name of black beans. Also keep the same term sprouted or germinated throughout the manuscript. 

The abbreviations of the samples are not listed with their explanation. 

L100: Please report the reason (microbial growth blocking?) for soaking in 10% sodium hypochlorite solution for 5 minutes. 

L111: Report briefly the extraction procedure

L115-125: Too much details. Kheldahl method is an AOAC standard method. Please report the code of the method and brief information. What ratio of N/protein ratio was used?

Protein secondary structure section lacks of discussion with previous relative findings. Also there could be a dicussion to associate FTIR findings with other findings of the study, solubility, in vitro digestion emulsifying or foaming ability etc. 

Fig 5B: The figure should not contain contact lines between the dots.

Author Response

Dear Reviewer :

We feel great thanks for your professional review work on our article. According to your nice suggestions, we have made corrections to our previous manuscript, and the specific concerns have been numbered. Our response is given in normal font and changes to the manuscript are given in the blue text.

Q1: Please mention the latin name of black beans. Also keep the same term sprouted or germinated throughout the manuscript.

A: We sincerely appreciate this valuable comment. In our resubmitted manuscript, the Latin name of black bean was added, and the terms of germination were unified. 

Q2: The abbreviations of the samples are not listed with their explanation.

A: Dear reviewer, we were really sorry for our careless mistakes. Thank you for your reminder. In our resubmitted manuscript, the abbreviations of the samples were listed.

Q3: L100: Please report the reason (microbial growth blocking?) for soaking in 10% sodium hypochlorite solution for 5 minutes.

A: Dear reviewer, sodium hypochlorite was used for simple disinfection of black bean seed skins 

Q4: L111: Report briefly the extraction procedure

A: We sincerely appreciate this valuable comment. In our resubmitted manuscript, the extraction method of germinated black bean protein isolate has been added.

Q5: L115-125: Too much details. Kheldahl method is an AOAC standard method. Please report the code of the method and brief information. What ratio of N/protein ratio was used?

A: We sincerely appreciate this valuable comment. Thank you for your reminder. In our resubmitted manuscript, this part has been modified.

Q6: Protein secondary structure section lacks of discussion with previous relative findings. Also there could be a dicussion to associate FTIR findings with other findings of the study, solubility, in vitro digestion emulsifying or foaming ability etc..

A: We sincerely appreciate this valuable comment. The related discussion has been added to the new manuscript. 

Q7: Fig 5B: The figure should not contain contact lines between the dots.

A: We sincerely appreciate this valuable comment. In our resubmitted manuscript, the picture has been altered.

We appreciate for Reviewers’ warm work earnestly, and hope the correction will meet with approval. Once again, thank you very much for your comments and suggestions..

Reviewer 3 Report

Comments and Suggestions for Authors

Manuscript ID:        foods-2837792

Title:                      Effects of germination on the structure, functional properties, 2 and in vitro digestibility of a black bean protein isolate

Abstract

The abstract is specific and concise. However, in my opinion should be shorter. Findings of whole study should be sum together in one sentence, instead of separately findings as authors proposed.

Keywords

In my opinion ‘functional nature’ should be replaced for ‘functional properties’, and ‘germinate’ for ‘germination’.

Introduction

The introduction is appropriate to the topic. Few corrections needed.

Line 52: Reference is needed. This sentence could be seen as controversial, especially by meat-focused scientists. Appropriate reference is necessary here.

Line 68: Prohealthy status should be mentioned ex. antidiabetic aspect of beans. There are some reference about it ex. 10.3390/ph15010065

In the Introduction, there should be mentioned about history of grain germination. 1-2 sentences would be enough.

Materials and Methods

Line 97 - Equipment needed.

Line 111 – mention Authors’ names before bracket [18]

Line 128 - by Ma et al. [19]

Line 133 - by Mokni et al. [20]

Line 193 - in Zhao et al. studies [27]

Line 211 - in Authors names [29]

Line 229- in Authors names [31]

The ‘Material & Methods’ is correct. Information about, how many repetitions in each analysis were conducted, should be added.

in vitro’ should be italic through the manuscript – please change all.

Results

The results were discussed adequately and sufficiently.

Figure 1 – add in footnote explanation about statistical analysis and letters a,b,c means

Table 1 – correct word into ‘Amino…’

Line 316 - seen from ours results (Figure 2)

3.3.2. – remove brackets when mention about result in table, Figure ex. line 338, 355, 367, 373, 391, 412, 437, 457, 469, 479, 493, 504, 519,

Font sizes are different in each table – unify according to journal requirements

Line 449 - by Yang et al. [68]

Figure 5, Figure 6 – table lines should be removed (invisible)

Line 530- of Bautista-Exposito et al/ [75] and Devi et al. [76].

The ‘Conclusions’ are correct, but weak. This part looks like shorter results, not conclusions. In my opinion should be rebuilt to get any more information about how results could be helpful for future investigations, and what other steps should be taken ?? What can we get additionally from these studies? Rebuild is needed.

Author Response

Dear Reviewer :

We feel great thanks for your professional review work on our article. As you are concerned, there are several problems that need to be addressed. According to your nice suggestions, we have made extensive corrections to our previous draft, and the specific concerns have been numbered. Our response is given in normal font and changes to the manuscript are given in the blue text.

Q1: The abstract is specific and concise. However, in my opinion should be shorter. Findings of whole study should be sum together in one sentence, instead of separately findings as authors proposed.

A: We sincerely thank the reviewer for carefully reading. In our resubmitted manuscript, the abstract was rewriting.

Q2: In my opinion ‘functional nature’ should be replaced for ‘functional properties’, and ‘germinate’ for ‘germination’

A: We sincerely appreciate the valuable comments. In our resubmitted manuscript, the wrong place was revised.

Q3: The introduction is appropriate to the topic. Few corrections needed.

Line 52: Reference is needed. This sentence could be seen as controversial, especially by meat-focused scientists. Appropriate reference is necessary here.

Line 68: Prohealthy status should be mentioned ex. antidiabetic aspect of beans. There are some reference about it ex. 10.3390/ph15010065IF: 4.6 Q2

A: We sincerely appreciate the valuable comments. Corresponding references have been added to the resubmitted manuscript.

Q4: In the Introduction, there should be mentioned about history of grain germination. 1-2 sentences would be enough

A: We sincerely appreciate the valuable comments, the sentences has been rewrite.

Q5: Materials and Methods

Line 97 - Equipment needed.

Line 111 – mention Authors’ names before bracket [18]

Line 128 - by Ma et al. [19]

Line 133 - by Mokni et al. [20]

Line 193 - in Zhao et al. studies [27]

Line 211 - in Authors names [29]

Line 229- in Authors names [31]

Line 449 - by Yang et al. [68]

Line 530- of Bautista-Exposito et al/ [75] and Devi et al. [76]

The ‘Material & Methods’ is correct. Information about, how many repetitions in each analysis were conducted, should be added.

‘in vitro’ should be italic through the manuscript – please change all

A: We sincerely thank the reviewer for carefully reading. As suggested by the reviewer, we have modified the above problems.

Q6: Figure 1 – add in footnote explanation about statistical analysis and letters a,b,c means

A: We are sorry for our carelessness. Thanks for your careful checks. A footnote for the figure has been added.

Q7: Table 1 – correct word into ‘Amino…’

A: We were really sorry for our careless mistakes. Thank you for your reminder. In our resubmitted manuscript, the wrong place was revised.

Q8: Line 316 - seen from ours results (Figure 2)

3.3.2. – remove brackets when mention about result in table, Figure ex. line 338, 355, 367, 373, 391, 412, 437, 457, 469, 479, 493, 504, 519

A: Thanks for your suggestion, brackets had been removed from the new manuscript.

Q9: Font sizes are different in each table – unify according to journal requirements

A: Thanks for your suggestion. According to the content of the form, we have unified the font size of the full table.

Q10: Figure 5, Figure 6 – table lines should be removed (invisible)

A: We were really sorry for our careless mistakes. Thank you for your reminder. In our resubmitted manuscript, the wrong place is revised.

Q11: The ‘Conclusions’ are correct, but weak. This part looks like shorter results, not conclusions. In my opinion should be rebuilt to get any more information about how results could be helpful for future investigations, and what other steps should be taken ?? What can we get additionally from these studies? Rebuild is needed.

A: We think this is an excellent suggestion. We have already re-written this part according to the Reviewer’s suggestion.

We tried our best to improve the manuscript and made some changes marked in blue in revised paper which will not influence the content and framework of the paper. We appreciate for Reviewers’ warm work earnestly, and hope the correction will meet with approval. Once again, thank you very much for your comments and suggestions.

Round 2

Reviewer 2 Report

Comments and Suggestions for Authors

Manuscirpt has been improved.

Author Response

Dear Editor :

We checked the references in the manuscript and revised line 197 to read: Wang et al. studies [29].

We tried our best to improve the manuscript and made some changes marked in blue in revised paper which will not influence the content and framework of the paper. We appreciate for your warm work earnestly, and hope the correction will meet with approval. Once again, thank you very much for your comments and suggestions.